# AGENT-CONTROLLER REPRESENTATIONS: PRINCIPLED OFFLINE RL WITH RICH EXOGENOUS INFORMATION

## ABSTRACT

Learning to control an agent from data collected offline in a rich pixel-based visual observation space is vital for real-world applications of reinforcement learning (RL). A major challenge in this setting is the presence of input information that is hard to model and irrelevant to controlling the agent. This problem has been approached by the theoretical RL community through the lens of *exogenous information*, i.e, any control-irrelevant information contained in observations. For example, a robot navigating in busy streets needs to ignore irrelevant information, such as other people walking in the background, textures of objects, or birds in the sky. In this paper, we focus on the setting with visually detailed exogenous information, and introduce new offline RL benchmarks offering the ability to study this problem. We find that contemporary representation learning techniques can fail on datasets where the noise is a complex and time dependent process, which is prevalent in practical applications. To address these, we propose to use multi-step inverse models, which have seen a great deal of interest in the RL theory community, to learn Agent-Controller Representations for Offline-RL (ACRO). Despite being simple and requiring no reward, we show theoretically and empirically that the representation created by this objective greatly outperforms baselines.

## 1 INTRODUCTION

Effective real-world applications of reinforcement learning or sequential decision-making must cope with exogenous information in sensory data. For example, visual datasets of a robot or car navigating in busy city streets might contain information such as advertisement billboards, birds in the sky or other people crossing the road walks. Parts of the observation (such as birds in the sky) are irrelevant for controlling the agent, while other parts (such as people crossing along the navigation route) are extremely relevant. How can we effectively learn a representation of the world which extracts just the information relevant for controlling the agent while ignoring irrelevant information?

Real world tasks are often more easily solved with fixed offline datasets since operating from offline data enables thorough testing before deployment which can ensure safety, reliability, and quality in the deployed policy (Lange et al., 2012; Ebert et al., 2018; Kumar et al., 2019; Jaques et al., 2019; Levine et al., 2020). The Offline-RL setting also eliminates the need to address exploration and planning which comes into play during data collection.[1] Although approaches from representation learning have been studied in the online-RL case, yielding improvements, exogenous information has proved to be empirically challenging. A benchmark for learning from offline pixel-based data (Lu et al., 2022a) formalizes this challenge empirically. Combining these challenges, is it possible to learn distraction-invariant representations with rich observations in offline RL?

Approaches for discovering small tabular-MDPs ($\leq 500$ discrete latent states) or linear control problems invariant to exogenous information have been introduced (Dieterich et al., 2018; Efroni et al., 2021; 2022b;a; Lamb et al., 2022) before. However, the planning and exploration techniques in these algorithms are difficult to scale. A key insight that Efroni et al. (2021); Lamb et al. (2022) uncovered is the usefulness of multi-step action prediction for learning exogenous-invariant representation.

---

[1]This elimination however can make offline RL more difficult if the wrong data is collected.

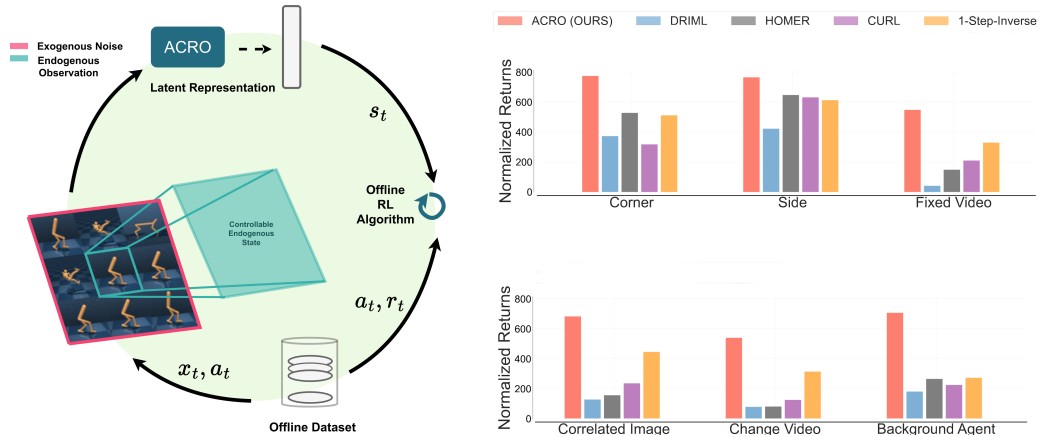

Figure 1: **Left: Representation Learning for Visual Offline RL in Presence of Exogenous Information**. We propose ACRO, that recovers the controller latent representations from visual data which includes uncontrollable irrelevant information, such as observations of other agents acting in the same environment. **Right: Results Summary**. ACRO learns to ignore the observations of task irrelevant agents, while baselines tend to capture such exogenous information. We use different offline datasets with varying levels of exogenous information (Section 4) and find that baseline methods consistently under-perform w.r.t. ACRO, as is supported by our theoretical analysis.

Following these, we propose to learn *Agent-Controller Representations for Offline-RL (ACRO)* using multi-step inverse models, which predict actions given current and future observations as in Figure 2. ACRO avoids the problem of learning distractors, because they are not predictive of the agent's actions. This property even holds for temporally-correlated exogenous information. At the same time, multi-step inverse models capture all the information that is sufficient for controlling the agent (Efroni et al., 2021; Lamb et al., 2022), which we refer to as the agent-controller representation. ACRO is learned in an entirely reward-free fashion. Our first contribution is to show that ACRO outperforms all current baselines on datasets from policies of varying quality and stochasticity. Figure 1 gives an illustration of ACRO, with a summary of our experimental findings.

A second core contribution of this work is to develop and release several new benchmarks for offline-RL designed to have especially challenging exogenous information. In particular, we focus on *diverse temporally-correlated* exogenous information, with datasets where (1) every episode has a different video playing in the background, (2) the same STL-10 image is placed to the side or corner of the observation throughout the episode, and (3) the observation consists of views of nine independent agents but the actions only control one of them (see Fig. 1). Task (3) is particularly challenging because which agent is controllable must be learned from data.

Finally, we also introduce a new theoretical analysis (Section 3) which explores the connection between exogenous noise in the learned representation and the success of Offline-RL. In particular, we show that Bellman completeness is achieved from the agent-controller representation of ACRO while representations which include exogenous noise may not verify it. Bellman completeness has been previously shown to be a sufficient condition for the convergence of offline RL methods based on Bellman error minimzation (Munos, 2003; Munos & Szepesvári, 2008; Antos et al., 2008).

## 2 ACRO: Agent-Controller Representations for Offline-RL

### 2.1 Preliminaries

We consider a Markov Decision Process (MDP) setting for modeling systems with both relevant and irrelevant components (also referred as exogenous block MDP in Efroni et al. (2021)). This MDP consists of a set of observations, $\mathcal{X}$; a set of latent states, $\mathcal{Z}$; a set of actions, $\mathcal{A}$; a transition distribution, $T(z' \mid z, a)$; an emission distribution $q(x \mid z)$; a reward function $R : \mathcal{X} \times \mathcal{A} \to \mathbb{R}$; and a start state distribution $\mu_0(z)$. We also assume that the support of the emission distributions of any two latent states are disjoint. The latent state is decoupled into two parts $z = (s, e)$ where $s \in \mathcal{S}$ is the agent-controller state and $e \in \mathcal{E}$ is the exogenous state. For $z, z' \in \mathcal{Z}, a \in \mathcal{A}$ the transition

function is decoupled as $T(z' \mid z, a) = T(s' \mid s, a)T_e(e' \mid e)$, and the reward only depends on $(s, a)$. These definitions imply that there exist mappings $\phi_\star : \mathcal{X} \to \mathcal{S}$ and $\phi_{\star,e} : \mathcal{X} \to \mathcal{E}$ from observations to the corresponding controller and exogenous latent states. The agent interacts with the environment, generating a latent state, observation and action sequence, $(z_1, x_1, a_1, z_2, x_2, a_2, \cdots,)$ where $z_1 \sim \mu(\cdot)$ and $x_t \sim q(\cdot \mid z_t)$. The agent does not observe the latent states $(z_1, z_2, \cdots)$, instead receiving only the observations $(x_1, x_2, \cdots)$. The agent chooses actions using a policy distribution $\pi(a \mid x)$. A policy is an *exo-free policy* if it is not a function of the exogenous noise. Formally, for any $x_1$ and $x_2$, if $\phi_\star(x_1) = \phi_\star(x_2)$, then $\pi(\cdot \mid x_1) = \pi(\cdot \mid x_2)$.

## 2.2 PROPOSED METHOD

We consider learning representations from an offline dataset $\mathcal{D} = (\mathcal{X}, \mathcal{A})$ consisting of sequences of N observations $\mathcal{X} = (x_1, x_2, x_3, ..., x_N)$ and the corresponding actions $\mathcal{A} = (a_1, a_2, a_3, ..., a_N)$. We are in the rich-observation setting, *i.e.*, observation $x_t \in \mathbb{R}^m$ is sufficient to decode $z_t$. Our focus is on pre-training an encoder $\phi : \mathbb{R}^m \to \mathbb{R}^d$ on $\mathcal{D}$ such that the frozen representation $s_t = \phi(x_t)$ is suitable for offline policy optimization.

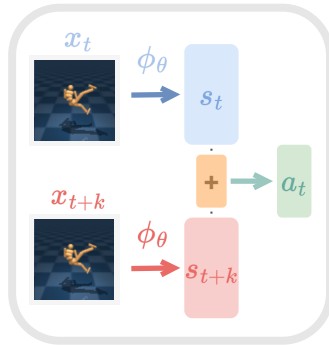

To learn representations that discard exogenous information, we leverage prior works from the theoretical RL community and train a multi-step inverse action prediction model, which captures long-range dependencies thanks to its conditioning on distant future observations. This leads to the ACRO objective, namely to predict the action conditioning on $\phi(x_t)$ and $\phi(x_{t+k})$. Note that even though we are conditioning on the future observation, we only predict the first action instead of the sequence of actions up to the k-th timestep, as the former is easier to learn.

Figure 2: **ACRO** is a multi-step inverse model that predicts the first action conditioned on the current state and the future state. **+** denotes concatenation.

Our proposed method, which we call *Agent-Controller Representations for Offline-RL* (ACRO), optimizes the following objective based on a multi-step inverse model:

$$\phi_\star \in \arg\max_{\phi \in \Phi} \mathbb{E}_{t \sim U(0,N)} \mathbb{E}_{k \sim U(0,K)} \log\left(\mathbb{P}(a_t \mid \phi(x_t), \phi(x_{t+k}))\right). \tag{1}$$

This approach is motivated by two desiderata: (i) ignoring exogenous information and (ii) capturing the latent state that is necessary for control. The following invariance lemma (Lemma 1, see Appendix B for proof) states that optimal action predictor models can be obtained without dependence on exogenous noise, when the data-collection policy is assumed not to depend on it either.

**Lemma 1** (Invariance Lemma: Multi-step inverse model is invariant to exogenous information, Lamb et al. (2022)). *For any exo-free policy $\pi : \mathcal{X} \to \mathcal{A}$, for all $a_t \in \mathcal{A}$, and $(x_t, x_{t+k}) \in$ supp $\mathbb{P}_\pi(X_t, X_{t+k})$:*

$$\mathbb{P}_\pi(a_t \mid x_t, x_{t+k}) = \mathbb{P}_\pi(a_t \mid \phi_\star(x_t), \phi_\star(x_{t+k})) \tag{2}$$

At the same time, prior works have shown that single step inverse models of action prediction can fail to capture the full controller latent states (Efroni et al., 2021; Lamb et al., 2022; Hutter & Hansen, 2022). One type of counter-example for single-step inverse models stems from a failure to capture long-range dependencies. For example, in an empty gridworld, a pair of positions which are two or more spaces apart can be mapped to the same representation without increasing the loss of a one-step inverse model. Another simple counter-example involves a problem where the last action the agent took is recorded in the observation, in which case the encoder can simply retrieve that action directly while ignoring all other information. The use of multi-step inverse models resolves both of these counter-examples and is able to learn the full agent-controller state (Lamb et al., 2022). A detailed related work discussion is provided in the Appendix Section G.

We emphasize here that even though inverse models of action prediction have appeared in past literature (as discussed in related works), they are often proposed for the purposes of exploration and

reward bonus. In contrast, we propose to learn the multi-step inverse model to explicitly uncover a representation that contains only the controller, endogenous part of the state. Recently, Lamb et al. (2022) proposed a multi-step inverse model where the learnt representation $\phi(\cdot)$ is regularized, so that $\phi(\cdot)$ discards irrelevant details from observations $x$. This was accomplished by using vector-quantization on the encoder's output, forcing discrete latent states to be learnt for constructing a tabular MDP for latent recovery. In contrast, ACRO learns the continuous endogenous latent state, without a bottleneck, and the learnt pre-trained representation $\phi(\cdot)$ is later used for policy optimization in offline RL. More details of our algorithm are discussed in Appendix E.

## 3 BENEFITS OF EXOGENOUS INVARIANT REPRESENTATION IN OFFLINE RL

Due to its importance to practical applications, the offline RL setting has been extensively studied by the theoretical community. The majority of provable value-based offline RL algorithms follow a Bellman error minimization approach (Munos, 2003; Munos & Szepesvári, 2008; Antos et al., 2008), in line with the techniques used in practice. The common representational assumptions needed to derive these results are: (A1) the function class contains the optimal Q function (realizability), (A2) the data distribution is sufficiently diverse (concentrability), and (A3) Bellman completeness (Munos & Szepesvári, 2008). This last condition is the most subtle one, it states that the function class can properly represent the Bellman backup of any function it contains.

**Definition 2** (Bellman Completeness). *We say that $\mathcal{F}$ is Bellman complete if it is closed under the Bellman operator. For any $f \in \mathcal{F}$, it holds that $\mathcal{T}f \in \mathcal{F}$, where $(\mathcal{T}f)(x, a) \equiv R(x, a) + \mathbb{E}_{x' \sim T(x'|x,a)}[\max_{a'} f(x', a')]$ for all $(x, a) \in \mathcal{X} \times \mathcal{A}$.*

Chen & Jiang (2019) conjectured that (A1) and (A2) alone are not sufficient for sample efficient offline RL, and, recently, Foster et al. (2021) established a lower bound proving this claim. Thus, the representational requirements needed for offline RL are more intricate than in supervised learning.

With these observations in mind, we highlight a key advantage of the agent-controller representation $\phi_\star$ relatively to other representations in the offline RL setting. Namely, one can construct a Bellman complete function class on top of $\phi_\star$, while some representations that include exogenous information provably violate the Bellman completeness property. To formalize these claims, we denote by $\mathcal{Q}_{\mathcal{S}} = \{(s, a) \mapsto [0, 1] : (s, a) \in \mathcal{S} \times \mathcal{A}\}$ the set of Q-functions defined over $\mathcal{S}$, and for a given representation $\phi$, we let $\mathcal{F}(\phi) = \{(s, a) \mapsto Q(\phi(s), a) : Q \in \mathcal{Q}_{\mathcal{S}}, (s, a) \in \mathcal{S} \times \mathcal{A}\}$ denote the set of Q-functions defined on top of $\phi$. The following proposition states that the Agent-Controller representation leads to a Bellman complete function class (all proofs in Appendix D.1/ Appendix D.2).

**Proposition 3** (ACRO Representation is Bellman Complete). *$\mathcal{F}(\phi_\star)$ is Bellman complete.*

Next, we show that there exists a representation strictly more expressive than ACRO (*i.e.*, one that includes exogenous information) which, surprisingly, violates the Bellman completeness property.

**Proposition 4** (Exogenous Information May Violate Bellman Completeness). *There exists $\phi$ which is a refinement[2] of $\phi_\star$ such that $\mathcal{F}(\phi)$ is not Bellman complete.*

This proposition implies that exogenous information being included in the representation may break the Bellman completeness assumption, which is a requirement for establishing the convergence of offline RL algorithms based on Bellman error minimization. From this perspective, additional information in the representation may deteriorate the performance of offline RL. Conversely, a coarser representation may trivially violate the realizability assumption A1: such a representation may merge states on which the optimal Q-function differs, preventing it from being realized.

Together, these observations motivate the experimental pipeline used this work: learn the agent-controller representation by optimizing Equation 1, then perform offline RL on top of it. In doing so, we obtain a representation that is sufficient for optimal performance, and yet filters the exogenous information which can (i) be impossible to exactly model, and (ii) hurt the offline RL performance.

## 4 EXPERIMENTS: OFFLINE RL WITH EXOGENOUS INFORMATION

This section provides extensive analysis of representation learning from visual offline data under rich exogenous information (Figure 4). Our experiments aim to understand the effect of exogenous

---

[2]Let $\mathcal{X}$ be a finite set of elements. Given a partition $P$ of $\mathcal{X}$ let its induced equivalence relation be denoted by $\sim_P$. A partition $P_1$ is finer than $P_2$ if for any $x_1, x_2 \in \mathcal{X}$ such that $x_1 \sim_{P_1} x_2$ it also holds that $x_1 \sim_{P_2} x_2$.

information and if ACRO can truly learn the agent controller state and thus improve performance in visual offline RL. **Due to space constraints, we defer all our experiment details and results to the Appendix Section A and only include method ablations and reconstructions in this section.**

### 4.1 Method Ablations and Analysis

We compare ACRO with three of its variations, 1) when $k = 1$, i.e. a standard one-step inverse model; 2) with $\mathbf{x}_{t+k}$ not provided as input, i.e. simply training the representation with a behavior cloning loss; and 3) when $m(k)$ the timestep embedding, is additionally provided as input. Ablations are shown over three different policies: random, medium-replay and expert in Table 1.

For both random and medium-replay policies, $k = 1$ leads to similar results than when $k$ is randomly chosen from 1 to 15. ACRO performs much better under an expert policy. We conjecture that the benefits of larger $k$ can only be realized when the policy is of a high enough quality to preserve information over long time horizons. Additionally training a behavior cloning loss performs similarly to ACRO for the medium-replay and expert datasets. However, when the actions come from a random policy, ACRO performs much better while the behavior cloning ablation collapses completely. This result is analyzed theoretically in Appendix C, which shows that ACRO is equivalent to behavior cloning under

Table 1: **Ablations** for different policies. The highlighted cells indicate where each variant fails to match ACRO's performance, hence showing that each component of ACRO is essential for consistently good performance. 5 seeds and std. dev. reported.

| Environment | Random | Medium-Replay | Expert |
|---|---|---|---|
| ACRO | $82.9 \pm 5.5$ | $228.8 \pm 50.1$ | $525.8 \pm 89.0$ |
| K=1 | $94.7 \pm 7.9$ | $241.0 \pm 9.9$ | $187.5 \pm 33.8$ |
| Only $x_t$ | $0.5 \pm 0.1$ | $229.4 \pm 64.7$ | $496.8 \pm 100.2$ |
| With $k$ | $43.1 \pm 49.5$ | $251.8 \pm 15.3$ | $302.2 \pm 29.1$ |

a deterministic and fixed expert policy, but should be much better otherwise. Adding a $k$ embedding generally degrades performance, although the effect is inconsistent. These results suggest that ACRO is a more well rounded and robust objective than other variants.

**Visualizing Reconstructions**. Having learnt a representation, we can train a decoder over it to minimize the reconstruction loss given the original observation. Such reconstructions would therefore measure how much information in the original observation is preserved in the representation, and thus act as a metric for evaluating the quality of representations. We compare such reconstructions in Figure 3 for the cheetah domain where the exogenous noise comes from a video playing in the background. Notably, ACRO is able to remove most background information while keeping the relevant body pose information intact. On the other hand, DRIML performs contrastive comparisons between states in a given trajectory and is not able to remove exogenous information quite as well. DRQ is able to remove exogenous noise but is unable to learn the controller state.

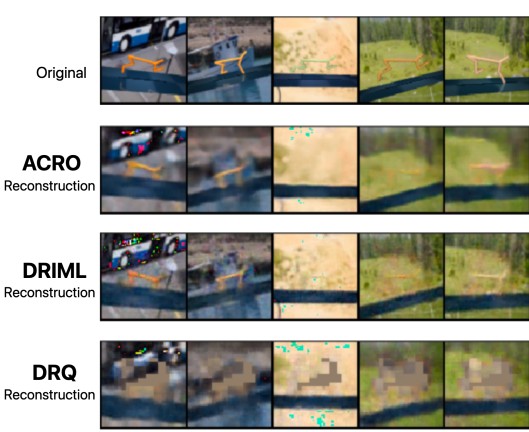

Figure 3: **Reconstructions** from a decoder with a static background image per episode: **Top-Bottom**: Original, ACRO, DRIML, DRQ.

## 5 Discussion

In this work, we introduced offline RL datasets with varying difficulties of exogenous information in the observations. Our results show that existing representation learning methods can significantly drop in performance for certain types of exogenous noise. We presented ACRO, a pre-training objective for offline RL based on a multi-step inverse prediction model, and showed it is far more robust to exogenous information, both theoretically and empirically. We hope this work will drive future interests in offline RL under different definitions of exogenous information.

## 6 REPRODUCIBILITY STATEMENT

The experiment details and use of different exogenous offline datasets are discussed in section 4. We provide further architecture and algorithm details in appendix E. For implementation, we use the open source codebase from the visual D4RL benchmark (Lu et al., 2022a) and implement ACRO and other baseline representation objectives using the same structure and pipeline from the codebase. All the representation learning objectives use the same encoder architecture and optimization specifications. In the main draft, we provide normalized performance plots for comparison, where we average across different types of datasets (expert, medium, medium-expert); and we provide individual performance plots for comparison in the appendix. We also provide code for our implementation along with supplementary materials. The use of different exogenous offline datasets hopefully introduces new benchmarks to be considered in the offline RL community, and we plan to release these benchmark datasets for future use. The proof for propositions and lemmas are included in details in the appendix.

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

# Appendix

## A  EXPERIMENTS

To this end, we evaluate ACRO against several state of the art representation learning baselines across two axes of added exogenous information: *Temporal Correlation* and *Diversity*, hence characterizing the level of difficulty systematically. We find that under exogenous information in offline RL, the performance of several state of the art representation learning objectives can degrade dramatically.

Two particular challenges in the datasets we explore are the temporal correlation and diversity in the exogenous noise. *Temporal Correlation:* Exogenous noise which lacks temporal correlation (time-independent noise) is relatively easy to filter out in the representation, especially in tasks where the agent-controller latent state has strong temporal correlation. *Diversity:* Similarly for the other axis, if exogenous noise is more diverse, it has a greater impact on the complexity of the subsequently learned representation. For example, if there are only two possible distracting background images, in the worst case the cardinality of a discrete representation is only doubled. On the other hand if there are thousands of possible distracting background images, then the effect on the complexity of representation would be far greater. We primarily categorize our novel visual offline datasets into *three categories* (Figure 4 in appendix provides observations under different exogenous distractors):

- **EASY-EXO**. Exogenous noise with low-diversity and no time correlation. **a)** Visual offline datasets from v-d4rl benchmark (Lu et al., 2022b) without any background distractors; **b)** Distractor setting (Lu et al., 2022a) with a single fixed exogenous image in the background.

- **MEDIUM-EXO**. Exogenous noise with either low-diversity or simple time-correlation. **a)** Exogenous image placed in the corner of agent observations, changes per episode; **b)** Exogenous image placed on the side of agent observations, changes per episode; **c)** A single fixed exogenous video playing in the background.

- **HARD-EXO**. Exogenous noise with both high-diversity and rich temporal correlation. **a)** Exogenous image in the background which changes per episode; **b)** Exogenous video in the background which changes per episode; **c)** Exogenous observations of nine agents placed in a grid, but the actions only control one of the agents (see Figure 1).

**Experiment Setup**. We provide details of each EXOGENOUS DATASETS in Appendix H.1, along with descriptions for the data collection process in Appendix I. Following Fu et al. (2020); Lu et al. (2022b), we release these datasets for future use by the RL community. All experiments involve pre-training the representation, and then freezing it for use in an offline RL algorithm. We use TD3 + BC as the downstream RL algorithm, along with data augmentations (Kostrikov et al., 2020), a combination which has been shown to be a reasonable baseline for visual offline RL (Lu et al., 2022b). Experiment setup and implementation details are discussed in Appendix E.

**Baselines**. We compare *five* other baselines, which are standard for learning representations of visual data. The baselines we consider are: (i) two temporal contrastive learning methods, DRIML (Mazoure et al., 2020) and HOMER (Misra et al., 2020); (ii) a data augmentation method, DRQ Kostrikov et al. (2020), and a spatial contrastive approach,

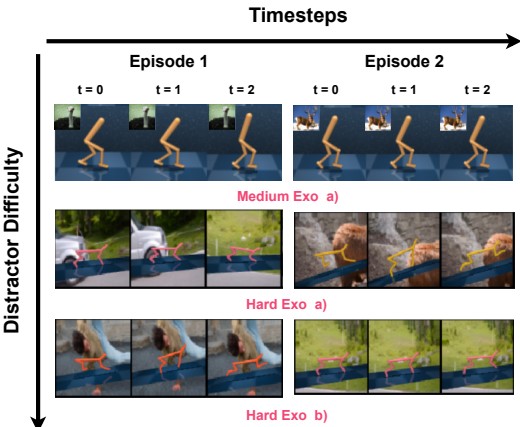

Figure 4: **Examples** of Different Categories of Exogenous Information. Further details of different exogenous information in offline datasets, with visual examples, are provided in Appendix H.1.

CURL Laskin et al. (2020); and (iii) inverse dynamics model learning, *i.e.*, 1-step inverse action

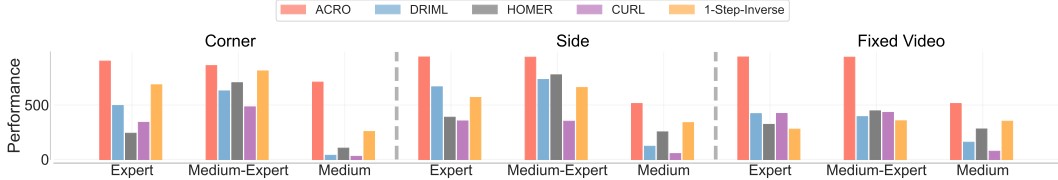

Figure 5: **MEDIUM-EXO Results**. Performance comparison of ACRO with several other baselines, with varying levels of exogenous information settings, either from STL10 dataset (Coates et al., 2011) or fixed video distractors in background during offline data collection.

prediction (Pathak et al., 2017). We do not consider baselines such as SPR (Schwarzer et al., 2020) and SGI (Schwarzer et al., 2021) which work well on the ALE Atari100K benchmark but not on continuous control benchmarks (Tomar et al., 2021). We also include preliminary Atari results in Appendix Section F where the representations are pre-trained using ACRO and used over a Decision Transformer architecture (Chen et al., 2021b).

## A.1 EASY-EXOGENOUS INFORMATION OFFLINE DATASETS

Table 2 summarizes results from the v-d4rl benchmark with visual offline data (Lu et al., 2022a). We label this as EASY-EXO since the dataset only contains a blank background without any additional exogenous noise being added. We find that ACRO learns a good agent-controller latent representation from pixel data with no apparent noise in observations, and can lead to effective performance improvements through pre-training representations. Extending results of EASY-EXO with static uncorrelated image background distractors from the v-d4rl benchmark, we see that the performance significantly decreases for all methods, while ACRO can strongly outperform all baselines, with the smallest drop in performance. Figure 6 shows normalized results across two different datasets and domains from v-d4rl. The distractors in this case belong to varying degree of shifts in the data distribution, according to (Lu et al., 2022a).

## A.2 MEDIUM-EXOGENOUS INFORMATION OFFLINE DATASETS

Figure 5 shows normalized results across three domains (cheetah-run, walker-walk, humanoid-walk) for the MEDIUM-EXO setting. Among these, the fixed background video is the hardest task. As expected, most methods underperform on data collected from a medium policy, compared to medium-expert and expert policies. However, ACRO consistently outperforms all methods across all datasets and distractor settings. Note the high variability in performance of baselines when changing the type of exogenous information (from corner, to side, to fixed video). Conversely, with the exception of Corner + Medium policy, ACRO performs similarly for all three settings. This suggests that baseline methods learn representations that are affected by the exogenous information, while ACRO remains rather impervious to it.

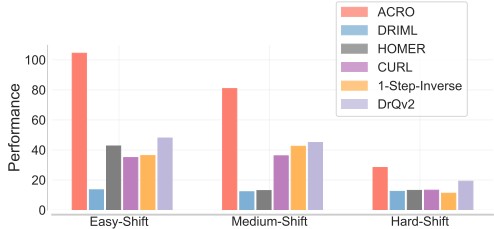

Figure 6: Normalized results across two domains from the v-d4rl distractor suite with varying levels of data shift severity. The easy, medium and hard categories are within the v-d4rl distractor suite of varying shift severity.

## A.3 HARD-EXOGENOUS INFORMATION OFFLINE DATASETS

With correlated exogenous noise in the form of either images or video, we observe that baseline representation objectives can be remarkably broken. Figure 7 shows normalized performance comparisons across different types of datasets (expert, medium-expert, medium) for three different types of HARD-EXO settings. Comparatively, ACRO can be more robust to the hard exogenous distractors, even though as the HARD-EXO types increase in difficulty, the maximum performance reached by all methods can degrade. Among the three HARD-EXO settings, changing video distractors in background during data collection seems to be the hardest, leading to performance drops for most methods. This suggests there is a strong correlation issue between the representation and the video

Table 2: **EASY-EXO**. Comparison of different representation methods on the standard v-d4rl benchmark, without additional exogenous information. ACRO consistently outperforms baseline methods in visual offline data. Performance plots in Appendix Figure 11. 10 seeds and std. dev. reported.

| ENVIRONMENT | DATASET | ACRO | DRIML | HOMER | DRQv2 | CURL | 1-STEP INVERSE |
|---|---|---|---|---|---|---|---|
| CHEETAH-RUN | Expert | $451.0 \pm 3.9$ | $330.2 \pm 2.9$ | $227.8 \pm 1.6$ | $256.9 \pm 2.2$ | $213.0 \pm 0.6$ | $239.9 \pm 0.4$ |
| | Medium-Expert | $466.0 \pm 3.2$ | $399.2 \pm 2.5$ | $390.7 \pm 1.3$ | $388.1 \pm 3.5$ | $328.4 \pm 2.1$ | $299.3 \pm 0.6$ |
| | Medium | $528.7 + 0.8$ | $508.5 \pm 0.7$ | $518.1 \pm 0.4$ | $488.3 \pm 0.5$ | $377.0 \pm 0.8$ | $400.3 \pm 0.4$ |
| | Medium-Replay | $416.9 \pm 0.9$ | $233.3 \pm 1.2$ | $333.2 \pm 1.2$ | $381.5 \pm 1.6$ | $279.4 \pm 1.8$ | $272.3 \pm .6$ |
| WALKER-WALK | Expert | $924.5 \pm 2.2$ | $485.10 \pm 4.9$ | $670.55 \pm 4.1$ | $888.6 \pm 6.0$ | $800.36 \pm 2.5$ | $831.5 \pm 3.4$ |
| | Medium-Expert | $914.6 \pm 1.8$ | $438.3 \pm 3.3$ | $774.5 \pm 2.5$ | $906.6 \pm 0.9$ | $724.6 \pm 4.5$ | $651.8 \pm 4.0$ |
| | Medium | $486.7 \pm 0.2$ | $469.4 \pm 0.5$ | $485.1 \pm 0.7$ | $425.6 \pm 1.6$ | $429.0 \pm 2.0$ | $389.4 \pm 1.1$ |
| | Medium-Replay | $277.8 \pm 0.5$ | $204.3 \pm 3.4$ | $318.9 \pm 4.0$ | $308.5 \pm 1.5$ | $234.8 \pm 2.4$ | $146.7 \pm 0.7$ |
| HUMANOID-WALK | Expert | $79.9 \pm 1.1$ | $17.5 \pm 0.1$ | $21.6 \pm 0.4$ | $34.1 \pm 0.3$ | $28.5 \pm 0.2$ | $25.4 \pm 0.1$ |
| | Medium-Expert | $142.4 \pm 1.2$ | $26.8 \pm 0.2$ | $31.8 \pm 0.1$ | $70.8 \pm 0.5$ | $63.2 \pm 0.9$ | $56.3 \pm 0.5$ |
| | Medium | $103.8 \pm 1.8$ | $35.1 \pm 0.3$ | $53.8 \pm 0.4$ | $96.4 \pm 0.9$ | $40.6 \pm 0.4$ | $46.7 \pm 0.1$ |
| | Medium-Replay | $197.8 \pm 0.5$ | $92.6 \pm 0.3$ | $102.7 \pm 0.6$ | $121.0 \pm 0.4$ | $77.8 \pm 0.8$ | $100.7 \pm 1.1$ |
| AVERAGE | | 415.8 | 270.0 | 327.4 | 363.9 | 299.7 | 288.4 |

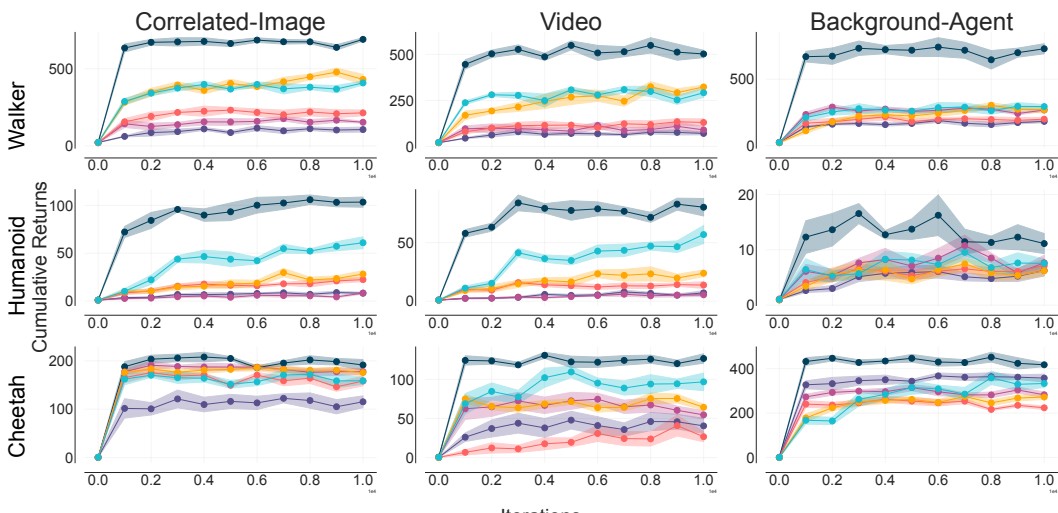

Figure 7: **HARD-EXO Results** Normalized performance across three datasets: medium, expert, and medium-expert. **First Column**. Time correlated exogenous distractor in background; **Second Column**. Video distractors that changes per episode in background; **Third Column**. Multiple background agent observations as distractors are placed in a grid of agent observation space

pixels, which breaks when the episode changes, hence leading to the worst scores across the three settings. However, ACRO remains comparatively robust and outperforms all baselines across all the HARD-EXO settings.

## B    INVARIANCE LEMMA PROOF

For any $k > 0$, consider $x, x' \in \mathcal{X}$ such that $x$ and $x'$ are separated by k steps. Both proofs first use bayes theorem, then apply the factorized transition dynamics, and then eliminate terms shared in the numerator and denominator. This proof is essentially the same as lemmas found in Lamb et al. (2022); Efroni et al. (2021), but is presented here for clarity.

**Lemma 1** states that the multi-step inverse model is invariant to exogenous noise. For any exo-free policy $\pi : \mathcal{X} \to \mathcal{A}$, for all $a_t \in \mathcal{A}$, and $(x_t, x_{t+k}) \in \text{supp } \mathbb{P}_\pi(X_t, X_{t+k})$:

$$\mathbb{P}_\pi(a_t \mid x_t, x_{t+k}) = \mathbb{P}_\pi(a_t \mid \phi_\star(x_t), \phi_\star(x_{t+k})) \tag{3}$$

*Proof.*

$$
\begin{aligned}
&\mathbb{P}_{\pi,\mu}(a \mid x', x) \\
&\overset{(a)}{=} \frac{\mathbb{P}_{\pi,\mu}(x' \mid x, a)\mathbb{P}_{\pi,\mu}(a \mid x)}{\sum_{a'} \mathbb{P}_{\pi,\mu}(x' \mid x, a')} \\
&\overset{(b)}{=} \frac{\mathbb{P}_{\pi,\mu}(x' \mid x, a)\pi(a \mid \phi_\star(x)))}{\sum_{a'} \mathbb{P}_{\pi,\mu}(x' \mid x, a')\pi(a' \mid \phi_\star(x))} \\
&\overset{(c)}{=} \frac{q(x' \mid \phi_\star(x'), \phi_{\star,e}(x'))\mathbb{P}_{\pi,\mu}(\phi_\star(x') \mid \phi_\star(x), a)\mathbb{P}_{\pi,\mu}(\phi_{\star,e}(x') \mid \phi_{\star,e}(x))\pi(a \mid \phi_\star(x))}{\sum_{a'} q(x' \mid \phi_\star(x'), \phi_{\star,e}(x'))\mathbb{P}_{\pi,\mu}(\phi_\star(x') \mid \phi_\star(x), a')\mathbb{P}_{\pi,\mu}(\phi_{\star,e}(x') \mid \phi_{\star,e}(x))\pi(a' \mid \phi_\star(x))} \\
&= \frac{\mathbb{P}_{\pi,\mu}(\phi_\star(x') \mid \phi_\star(x), a)\pi(a \mid \phi_\star(x))}{\sum_{a'} \mathbb{P}_{\pi,\mu}(\phi_\star(x') \mid \phi_\star(x), a')\pi(a' \mid \phi_\star(x))}.
\end{aligned}
$$

$\square$

Relation $(a)$ holds by Bayes' theorem. Relation $(b)$ holds by the assumption that $\pi$ is uniformly random (in the first proof) or exo-free (in the second proof). Relation $(c)$ holds by the factorization property. Thus, $\mathbb{P}_{\pi,\mu}(a \mid x', x) = \mathbb{P}_{\pi,\mu}(a \mid \phi_\star(x'), \phi_\star(x))$, and is constant upon changing the observation while fixing the agent controller state.

## C    CONNECTION BETWEEN ACRO AND BEHAVIOR CLONING

In the special case where all data is collected under a fixed deterministic, exogenous-free policy, ACRO and behavior cloning become equivalent. This case can still be non-trivial, if the start state of the episode is stochastic or if the environment dynamics are stochastic.

**Lemma 5.** *Under fixed, deterministic, and exo-free policy $\hat{\pi} : \mathcal{X} \to \mathcal{A}$, multi-step inverse model is equivalent to behavior cloning. For all $a_t \in \mathcal{A}$, and $(x_t, x_{t+k})$ such that $\mathbb{P}_{\hat{\pi}}(X_t = x_t, X_{t+k} = x_{t+k}) > 0$ we have:*

$$\mathbb{P}_{\hat{\pi}}(a_t \mid \phi_\star(x_t), \phi_\star(x_{t+k})) = \mathbb{P}_{\hat{\pi}}(a_t \mid \phi_\star(x_t)) \tag{4}$$

The proof of this claim is simply that behavior cloning is already able to predict actions perfectly in this special case, so there can be no benefit to conditioning on future observations.

*Proof.* By the assumption of the deterministic exo-free policy, we have that $\mathbb{P}_{\hat{\pi}}(a_t = \hat{a}(\phi_\star(x_t)) \mid \phi_\star(x_t)) = 1$ where $\hat{a} : \mathcal{S} \to A$ is a function mapping the latent state to the action.

Using bayes theorem we write:

$$\mathbb{P}_{\hat{\pi}}(a_t \mid \phi_\star(x_t), \phi_\star(x_{t+k})) = \frac{\mathbb{P}_{\hat{\pi}}(\phi_\star(x_{t+k} \mid \phi_\star(x_t), a_t)\mathbb{P}_{\hat{\pi}}(a_t \mid \phi_\star(x_t))}{\sum_{a''} \mathbb{P}_{\hat{\pi}}(\phi_\star(x_{t+k} \mid \phi_\star(x_t), a'')\mathbb{P}_{\hat{\pi}}(a'' \mid \phi_\star(x_t))} \tag{5}$$

For any examples in the dataset and for all $a' \in A$:

Case 1: $a' = \hat{a}(\phi_\star(x_t))$. It holds that

$$\mathbb{P}_{\hat{\pi}}(a_t \mid \phi_\star(x_t), \phi_\star(x_{t+k})) = \frac{\mathbb{P}_{\hat{\pi}}(\phi_\star(x_{t+k} \mid \phi_\star(x_t), a_t)\mathbb{P}_{\hat{\pi}}(a_t \mid \phi_\star(x_t))}{\sum_{a''} \mathbb{P}_{\hat{\pi}}(\phi_\star(x_{t+k} \mid \phi_\star(x_t), a'')\mathbb{P}_{\hat{\pi}}(a'' \mid \phi_\star(x_t))}$$

$$\mathbb{P}_{\hat{\pi}}(a_t \mid \phi_\star(x_t), \phi_\star(x_{t+k})) = \frac{\mathbb{P}_{\hat{\pi}}(\phi_\star(x_{t+k} \mid \phi_\star(x_t), a_t = a')}{\mathbb{P}_{\hat{\pi}}(\phi_\star(x_{t+k} \mid \phi_\star(x_t), a_t = a')}$$

$$\mathbb{P}_{\hat{\pi}}(a_t \mid \phi_\star(x_t), \phi_\star(x_{t+k})) = 1.$$

On the other hand, it holds that $\mathbb{P}_{\hat{\pi}}(a_t = a' \mid \phi_\star(x_t)) = 1$ since $a' = \hat{a}(\phi_\star(x_t))$. Hence, for this case, the claim holds true.

Case 2: $a' \neq \hat{a}(\phi_\star(x_t))$. It holds that

$$\mathbb{P}_{\hat{\pi}}(a_t \mid \phi_\star(x_t), \phi_\star(x_{t+k})) = \frac{\mathbb{P}_{\hat{\pi}}(\phi_\star(x_{t+k} \mid \phi_\star(x_t), a_t)\mathbb{P}_{\hat{\pi}}(a_t \mid \phi_\star(x_t))}{\sum_{a''} \mathbb{P}_{\hat{\pi}}(\phi_\star(x_{t+k} \mid \phi_\star(x_t), a'')\mathbb{P}_{\hat{\pi}}(a'' \mid \phi_\star(x_t))}$$

$$\mathbb{P}_{\hat{\pi}}(a_t \mid \phi_\star(x_t), \phi_\star(x_{t+k})) = \frac{0}{\mathbb{P}_{\hat{\pi}}(\phi_\star(x_{t+k} \mid \phi_\star(x_t), a_t = \hat{a}(\phi_\star(x_t)))}$$

$$\mathbb{P}_{\hat{\pi}}(a_t \mid \phi_\star(x_t), \phi_\star(x_{t+k})) = 0.$$

On the other hand, it holds that $\mathbb{P}_{\hat{\pi}}(a_t = a' \mid \phi_\star(x_t)) = 0$ since $a' \neq \hat{a}(\phi_\star(x_t))$. Hence, for this case the claim also holds true. This concludes the proof since the two distributions are equal in both cases. $\square$

# D BENEFITS OF EXOGENOUS INVARIANT REPRESENTATION IN OFFLINE RL

## D.1 PROOF OF PROPOSITION 3.

We need to show that for any $f \in \mathcal{F}(\phi_\star)$ and $x, a$, it holds that

$$R(x, a) + \mathbb{E}_{x' \sim T(\cdot | x, a)}[\max_{a'} f(x', a')] \tag{6}$$

is contained in $\mathcal{F}(\phi_\star)$. Since the reward is a function of the agent controller representation only, and since $f \in \mathcal{F}(\phi_\star)$, equation 6 can be written as:

$$R(x, a) + \mathbb{E}_{x' \sim T(\cdot | x, a)}[\max_{a'} f(x', a')]$$

$$= R(\phi_\star(x), a) + \mathbb{E}_{x' \sim T(\cdot | x, a)}[\max_{a'} f(\phi_\star(x'), a')]$$

$$= R(\phi_\star(x), a) + \mathbb{E}_{x' \sim T(\cdot | \phi_\star(x), a)}[\max_{a'} f(\phi_\star(x'), a')]. \tag{7}$$

The first relation holds since $f \in \mathcal{F}(\phi_\star)$ and by the assumption on the reward function (that it is a function of the endogenous states). The second relation holds by

$$\mathbb{E}_{x' \sim T(\cdot | x, a)}[\max_{a'} f(\phi_\star(x'), a')]$$

$$\overset{(a)}{=} \sum_{s', e'} \sum_{x' \in \text{supp} q(x' | s', e')} q(x' \mid \phi_\star(x'), \phi_{\star, e}(x'))T(s' \mid \phi_\star(x), a)T_e(e' \mid \phi_{\star, e}(x))f(s', a')$$

$$\overset{(b)}{=} \sum_{s'} T(s' \mid \phi_\star(x), a)f(s', a') \sum_{e'} T_e(e' \mid \phi_{\star, e}(x))$$

$$\overset{(c)}{=} \sum_{s'} T(s' \mid \phi_\star(x), a)f(s', a'),$$

where $(a)$ holds by the Ex-BMDP transition model assumption,

$$T(x' \mid x, a) = q(x' \mid \phi_\star(x'), \phi_{\star, e}(x'))T(\phi_\star(x') \mid \phi_\star(x), a)T_e(\phi_{\star, e}(x') \mid \phi_{\star, e}(x)),$$

$(b)$ and $(c)$ hold by marginalizing over $x'$ and $e'$. This establishes equation 7 and the proposition: the function $R(\phi_\star(x), a) + \mathbb{E}_{x' \sim T(\cdot | \phi_\star(x), a)}[\max_{a'} f(\phi_\star(x'), a')]$ is contained within $\mathcal{F}(\phi_\star)$ since it only depends on $\phi_\star$.

## D.2 PROOF OF PROPOSITION 4.

Consider an Ex-BMDP with one action $a$ where the agent controller representation is trivial and has a single fixed state (the agent has no ability to affect on the dynamics). We will establish a counter-example by constructing a tabular-MDP. Because tabular-MDP is a special case of a more general MDP with continuous states, this will also establish a counterexample for the more general non-tabular setting considered in the paper.

Let the observations and dynamics be given has follows. The observation is a 2-dimensional vector $x = (x(1), x(2))$ where $x(1), x(2) \in \{0, 1\}$. The dynamics is deterministic and its time evoluation is given as follows:

$$x_{t+1}(1) = x_t(1) \oplus x_t(2)$$
$$x_{t+1}(2) = x_t(2),$$

where $\oplus$ is the XOR operation. In this case, the transition model is given by $T(x' \mid x, a) = T(x' \mid x)$ and $\phi_\star = \{s_0\}$ where $s_0$ is a single state; since the observations are not controllable the controller representation maps all observations to a unique state. Further, assume that the reward function is $0$ for all observations.

Assume that $\phi(x) = (x_1)$, i.e., the representation ignores the second feature $x_2$. This representation is more refined than $\phi_\star$ since the latter maps all observations into the same state. Consider the tabular Q function class on top of this representation $\mathcal{Q}_{N=2}$, and consider $Q \in \mathcal{Q}_{N=2}$ given as follows

$$Q(x_1 = 1) = 1$$
$$Q(x_1 = 0) = 0.$$

We now show that $\mathcal{T}Q$ is not contained in $\mathcal{Q}_{N=2}$. According to the construction of the transition model, it holds that

$$(TQ)(x_1 = 1, x_2 = 1) = 0$$
$$(TQ)(x_1 = 0, x_2 = 1) = 1$$
$$(TQ)(x_1 = 1, x_2 = 0) = 1$$
$$(TQ)(x_1 = 0, x_2 = 0) = 0.$$

This function cannot be represented by a function from $\mathcal{Q}_{N=2}$; we cannot represent $(TQ)$ since it is not a mapping of the form $x_1 \to \mathbb{R}$ by the fact that, e.g.,

$$(TQ)(x_1 = 1, x_2 = 1) \neq (TQ)(x_1 = 1, x_2 = 0).$$

Meaning, it depends on the value of $x_2$.

## E   EXPERIMENT SETUP AND DETAILS

In this section, we describe our experiment setup in details. In all our experiments, comparing ACRO with other baseline representation objectives, we pre-train the representations for $100K$ pre-training steps. We also include results with end to end fine-tuning of representations, in the appendix, and find that the experimental comparisons across the different datasets we considered are consistent. Given pixel based visual offline data, we use a simple CNN+MLP architecture for encoding obeservations and predicting the ACRO actions. We also use cropping-based data augmentation as in DrQv2 while pre-training the representations for all methods. Specifically, the ACRO encoder uses 4-layers of convolutions, each with a kernel size of 3 and 32 channels. The original observation is of $84 \times 84 \times 9$, corresponding to a 3 channel-observation and a frame stacking of 3. The final encoder layer is an MLP which maps the convolutional output to a representation dimension of 256, giving the output $\phi(x)$. This is followed by a 2-layer MLP (hidden dim-256) that is used to predict the action given a 512 input corresponding to a concatenated $s_t$ and $s_{t+k}$ representations. For ACRO, we sample $k$ from 1 to 15 uniformly. We use ReLU non-linearity and ADAM for optimization all throughout.

For our experiments, we build off from the open source code base accompanying the v-d4rl benchmark (Lu et al., 2022a). We implement the pre-trained representation objectives in a model-free setting, where for the baseline offline RL algorithm we use **TD3 + BC** (Fujimoto & Gu, 2021b) since it achieves state of the art performance in raw state based offline RL benchmarks (Fu et al., 2020) and is a reasonably performing algorithm for visual offline RL as well Lu et al. (2022a). The policy improvement objective for the baseline **TD3 + BC** algorithm is thus:

$$\pi = \arg\max_{\pi} \mathbb{E}_{(\mathbf{s}_t, \mathbf{a}_t) \sim \mathcal{D}_{\text{env}}} \left[ \lambda Q(\mathbf{s}_t, \pi(\mathbf{s}_t)) - \left( \pi(\mathbf{s}_t) - \mathbf{a}_t \right)^2 \right] \tag{8}$$

where the critic $Q(s, \pi(s))$ is evaluated by a TD loss, and we use re-paramterized gradients through the critic for policy improvement step. For pixel based visual observations, recent work Lu et al. (2022a) proposed the **DrQ + BC** algorithm, which is essentially the TD3 + BC algorithm, except it additionally applies the data augmentations on pixel based inputs. In DrQ, data augmentation is applied only to the images sampled from the replay buffer, and not during the sample collection procedure. Given the pixel based control tasks, where the images are $84 \times 84$, DrQ pads each side by 4 pixels (repeating boundary pixels) and then selects a random $84 \times 84$ crop, yielding the original image, shifted by 4 pixels. This procedure is repeated every time an image is sampled from the replay buffer; and makes DrQ data augmentation quite effective based on the random shifts alone, without the need for any additional auxiliary losses. More concretely, denoting policy as $\pi_\theta$, the policy network is trained with the following loss :

$$L_\theta(\mathcal{D}) = -\mathbb{E}_{x_t \sim \mathcal{D}} \left[ Q_\psi(\mathbf{z}_t, \tilde{a}_t) \right] \tag{9}$$

where $\mathbf{z}_t = f_\epsilon(\text{aug}(x_t))$ is the encoded augmented visual observation, $\tilde{a}_t = \pi_\theta(\mathbf{z}_t) + \epsilon$ (action with clipped noise to smooth targets, $\epsilon \sim \text{clip}(\mathcal{N}(0, \sigma^2), -c, c)$). Overall loss for policy improvement :

$$\mathcal{L}_\theta(\mathcal{D}) = -\mathbb{E}_{x_t, \mathbf{a}_t \sim \mathcal{D}} \left[ \lambda Q_\psi(\mathbf{z}_t, \mathbf{a}_t) - (\pi_\theta(\mathbf{z}_t) - \mathbf{a}_t)^2 \right] \tag{10}$$

where DrQ passes the gradients of the critic to learn the encoder, and there are no separate or explicit representation losses other than the critic estimation, for training the encoder in DrQ. For all our experiments, in the EASY-EXO setting, we use datasets provided from v-d4rl benchmark (Lu et al., 2022a). For the MEDIUM-EXO and HARD-EXO settings, we follow the procedure from (Fu et al., 2020; Lu et al., 2022a) to collect new datasets using a soft actor-critic (SAC) policy, under different variations of exogenous information that we considered in this work. Details of the data collection procedure for the different settings are discussed in details below.

## F  ADDITIONAL EXPERIMENT RESULTS: ATARI MEDIUM-EXO

We also consider the setting of Atari. We build on the setup introduced in decision transformer (Chen et al., 2021b). While decision transformer focuses on framing the reinforcement learning problem as a sequence modeling problem, we mainly focus on learning representations which can learn to ignore exogenous noise. We use the same 4 games used in Chen et al. (2021b) - Pong, Qbert, Breakout, and Seaquest. We consider the MEDIUM-EXO setting where a different image is used in each episode and concatenated to the side of each observation. We add randomly sampled images from the CIFAR10 dataset (Krizhevsky & Hinton, 2009) as exogenous noise. Figure 8 shows an example observation from breakout with exogenous noise.

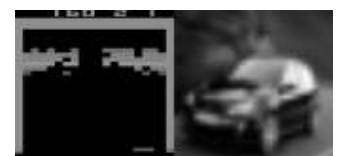

Figure 8: **Example** of an observation from Breakout with a CIFAR10 image on the side.

Decision Transformer uses the DQN-Replay dataset (Agarwal et al., 2020) for training. The model is trained using a sequence modeling objective to predict the next action given the past states, actions, and returns-to-go $\hat{R}_c = \sum_{c'=c}^{C} r_c$, where $c$ denotes the timesteps. This results in the following trajectory representation: $\tau = (\hat{R}_1, s_1, a_1, \hat{R}_2, s_2, a_2, \hat{R}_3, s_3, a_3, \dots)$, where $a_c$ denotes the actions and $s_c$ denotes the states. At test time, the start state $s_1$ and desired return $\hat{R}_1$ is fed into the model and it autoregressively generates the rest of the trajectory.

Decision Transformer uses a convolutional encoder to encode the observations. We first pretrain this encoder using the proposed ACRO objective. We use the 1-step inverse objective and DRIML as our baselines. After pretraining, we train the decision transformer using the sequence modeling objective keeping the encoder fixed. We present results in Table 3. We can see that ACRO outperforms both the baselines in all games further showing the effectiveness of the proposed approach.

Table 3: **Atari (MEDIUM-EXO)**. Here we compare ACRO to One-Step-Inverse model and DRIML on various games from the atari benchmark with exo-noise. We can see that ACRO consistently outperforms both the baselines. Results averaged across 5 seeds.

| Game | One-Step Inverse | DRIML | ACRO |
|---|---|---|---|
| Breakout | $3.8_{\pm 0.4}$ | $1.0_{\pm 0.0}$ | $20.6_{\pm 3.2}$ |
| Pong | $8.6_{\pm 3.2}$ | $-20.0_{\pm 0.0}$ | $11.8_{\pm 3.37}$ |
| Qbert | $536.2_{\pm 233.75}$ | $277.8_{\pm 46.24}$ | $657.4_{\pm 271.52}$ |
| Seaquest | $274.0_{\pm 29.61}$ | $94.4_{\pm 4.63}$ | $972.4_{\pm 136.09}$ |

**Hyperparameter Details**. We keep most of the hyperparameter details same as used in Chen et al. (2021b). They use episodes of fixed length during training - also referred to as the *context length*. We use a context length of 30 for Seaquest and Breakout and 50 for Pong and Qbert. Similar to Chen et al. (2021b), we consider one observation to be a stack of 4 atari frames. To implement the ACRO objective, we sample 8 different values for $k$ and calculate the objective for each value of $k$, obtaining the final loss by taking the sum across all the sampled values of $k$. We do not feed the embedding for $k$ in the MLP that predicts the action while computing the ACRO objective.

Table 4: **Overview of Properties** of prior works on representation learning in RL, in particular their robustness to exogenous information. The comparison to ACRO aims to be as generous as possible to the baselines. ✗ is used to indicate a known counterexample for a given property.

| Algorithms | TD3 (DrQ) | CURL | DRIML | DBC | AE | 1-Step Inverse | Behavior Cloning | BYOL Explore | **ACRO** (Ours) |
|---|---|---|---|---|---|---|---|---|---|
| Time-Ind. Exo-Invariant | ✓ | ✗ | ✓ | ✓ | ✗ | ✓ | ✓ | ✓ | ✓ |
| Reward Free | ✗ | ✓ | ✓ | ✗ | ✓ | ✓ | ✓ | ✓ | ✓ |
| Exogenous Invariant | ✗ | ✗ | ✗ | ✓ | ✗ | ✓ | ✓ | ? | ✓ |
| Non-Expert Policy | ✓ | ✓ | ✓ | ✓ | ✓ | ✓ | ✗ | ✓ | ✓ |
| Agent-Controller Rep. | ✓ | ✗ | ✓ | ✗ | ✓ | ✗ | ✓ | ✗ | ✓ |

# G  RELATED WORK

In Table 4, we list prior works and whether they verify various properties, in particular invariance to exogenous information. An extended discussion on related works is provided in Appendix G.

**Inverse Dynamics Models**. One-Step Inverse Models predict the action taken conditioned on the previous and resulting observations. This is invariant to exogenous noise but fails to capture the agent-controller latent state (Efroni et al., 2021), as previously discussed in Section 2.2. This can result from inability to capture long-range dependencies (Lamb et al., 2022) or could result from trivial prediction of actions using a dashboard displaying the last action taken, such as the brakelight which turns on after the break is applied on a car (De Haan et al., 2019). Behavior Cloning predicts actions given current state and may also condition on future returns. This is invariant to exogenous noise but can struggle with non-expert policies and generally fails to learn agent-controller latent state. Inverse models predicting sequences of actions, like GLAMOR (Paster et al., 2020) considers an online setting where they learn an action sequence as a sequential multi-step inverse model and rollout via random shooting and re-scoring, using both the inverse-model accuracies and an action-prior distribution. On the other hand, we learn a representation fully offline with a multi-step inverse model and then do policy optimization over the learnt representation, given fixed dataset.

**Contrastive Methods**. CURL (Augmentation Contrastive, Laskin et al. (2020)) learns a representation which is invariant to a class of data augmentations while being different across random example pairs. Depending on what augmentations and datasets are used, the learnt representations would generally learn exogenous noise and also fail to capture agent-controller latent states (which could be removed by some augmentations). HOMER and DRIML (Time Contrastive, Misra et al. (2020), Mazoure et al. (2020)) learns representations which can discriminate between adjacent observations in a rollout and pairs of random observations. This has been proven to not be invariant to exogenous information and neither can capture the agent-controller latent state (Efroni et al., 2021).

**Predictive Models**. Autoencoders learn to reconstruct an observation through a representation bottleneck. Generative modeling approaches usually capture all information in the input space which includes both exogenous noise and the agent-controller latent state (Hafner et al., 2019). Wang et al. (2022a;b) showed that a generative model of transition in the observation space can decompose the space into agent-controller state and exogenous information. While this does, in principle, eventually achieve an Agent-Controller representation, it comes at the cost of learning the exogenous representation and its dynamics before discarding the information. BYOL-EXPLORE (Guo et al., 2022) achieved impressive empirical results in online exploration by predicting future representations based on past representations and actions. While this approach can ignore exogenous information, there is no guarantee that it will do so, nor that it will learn the full agent-controller state.

**Reward-Dependent Methods**. DRQV2 (Kostrikov et al., 2020) learns a value function from offline tuples of observations, rewards, and actions. This could feasibly ignore exogenous noise given a suitable data-collection policy, but will not generally learn the full agent-controller latent state due to a heavy dependence on the reward structure. DBC (Bisimulation, Zhang et al. (2020)) learns representations which have similar values under a learned value function. In general, bisimulation is an overly restrictive state abstraction that fails to transfer to different tasks.

**RL with Exogenous Information**. Several prior works study the RL with exogenous information problem. In Dietterich et al. (2018); Efroni et al. (2022a;b) the authors consider specific represen-

tational assumptions on the underlying model, such as linear dynamics or factorized representation of the exogenous information in observations. Our work focuses on the rich observation setting where the representation itself should be learned. Efroni et al. (2021) propose a deterministic path planning algorithm for being invariant to exogenous noise. Unlike their approach which requires interaction with the environment using a tabular-MDP, ACRO is a purely offline algorithm. Lastly, the work of Lamb et al. (2022) suggests an endogenous latent state recovery algorithm through the use of a discretization bottleneck to construct a small tabular-MDP. In contrast, ACRO recovers the continuous counterpart of the endogenous latent state space directly, without the need to construct a tabular-MDP. Moreover, here we focus on reward optimization, and not only latent state discovery.

**Representation learning in Offline RL**. Representation learning offers an exciting avenue to address the demands of learning compact feature for state by incorporating the auxiliary task of the state feature within the learning task. Empirical studies on representation learning in Offline RL have been first addressed by Yang & Nachum (2021), which evaluate the ability of a broad set of representation learning objectives in the offline dataset and propose Attentive Contrastive Learning (ACL) to improve downstream policy performance. After that, Chen et al. (2021a) investigate whether the auxiliary representation learning objectives that broadly used in NLP or CV domains can help for imitation across different Offline RL tasks. Lu et al. (2022b) further explores the existing challenges for visual observation input with the Offline RL dataset, meanwhile providing simple modifications on several state-of-the-art Offline RL algorithms to establish a competitive baseline. Another branch of representation learning in Offline RL is theoretical side. Uehara et al. (2021) studies the representation learning in low-rank MDPs with Offline settings and proposes an algorithm that leverages pessimism to learn under a partial coverage condition, Nachum & Yang (2021) develops a representation objective that provably accelerate the sample-efficiency of downstream Offline RL tasks, Ghosh & Bellemare (2020) theoretically shows that the stability of the policy is tightly connected with the geometry of the transition matrix, which can provide stability conditions for algorithms that learn features from the transition matrix of a policy and rewards.

**Offline RL**. The predominant approach to train offline RL agent is regularizing the learned policy to be close to the behavior policy of the offline dataset. This can be implemented by generating the actions that similar to the dataset and restricting the output of the learned policy close to the generated actions (Fujimoto et al., 2019), penalizing the distance between the learned policy and the behavior of the dataset (Kumar et al., 2019; Zhang et al., 2021b), or introducing a pessimism term to regularize the Q function for avoiding high Q value of the out-of-distribution actions (Kumar et al., 2020; Buckman et al., 2021). Some approaches utilize BC as a reference for policy optimization with the baseline methods (Fujimoto & Gu, 2021a; Laroche et al., 2019; Nadjahi et al., 2019; Simão et al., 2020; Rajeswaran et al., 2018). Some other approaches improve the performance by measuring the uncertainty of the model's prediction (Yu et al., 2020; Kidambi et al., 2020; An et al., 2021).

## H    EXOGENOUS INFORMATION DATASETS

In this section, we provide a detailed summary of the different types of exogenous information based datasets as demonstrated in Figure 4. In Figure 9 we show samples from all of the datasets we explored. We further provide details for how each dataset is collected in Appendix I.

### H.1    DATASET DETAILS

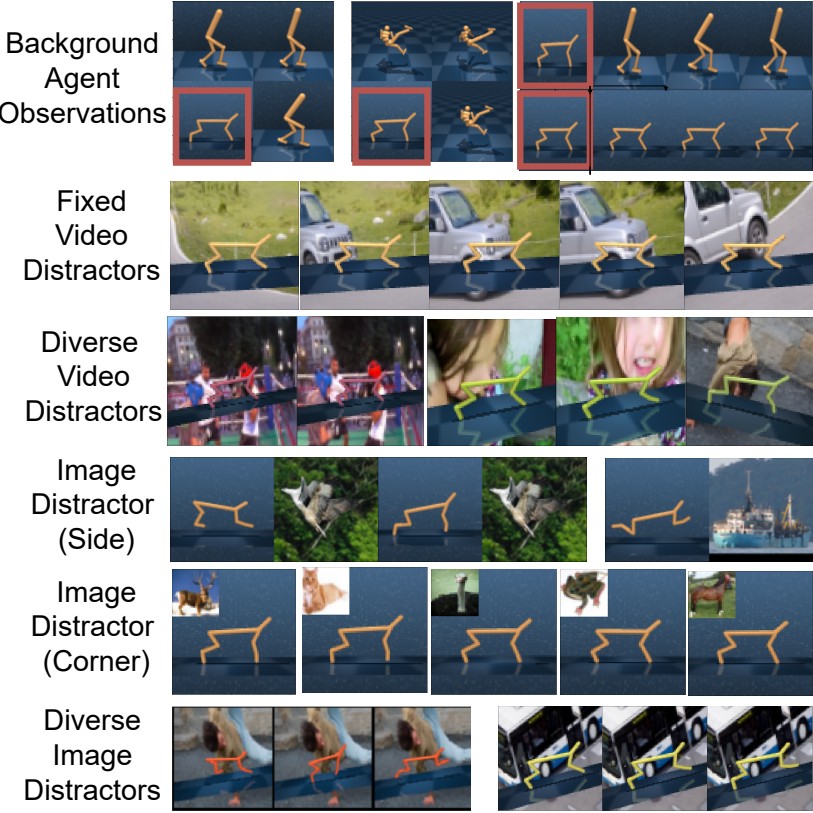

Figure 9: **Summary of Different Exogenous Information Offline Datasets**. In this work, we collected new datasets under different exogenous observations for offline RL. **Top to Bottom**: (a) **Background Agent Observations as Exogenous Information** (first row) showing endogenous controller agent (in red) and other exogenous agent observations taking random actions from other environments. (b) **Fixed and Changing Video Distractors** (second and third row respectively) where background video distractor changes per episode during offline data collection. (c) **Uncorrelated Image Distractors** (fourth and fifth row) showing that exogenous distactors can either be on the background, on the side or in the corner of the agent's observation space. (d) **Correlated Background Image Distractors** (sixth row) where background image distractor remains fixed per episode during data collection, and only changes per episode, to introduce time correlated exogenous image distractors. We find as the type of exogenous distractor becomes harder, from uncorrelated to correlated exogenous noise, the ability for baselines to learn good policies significantly breaks, as seen from the performance evaluations; whereas ACRO can still be robust to the exogenous information.

**Easy-Exogenous Information (EASY-EXO)**. A visual RL offline benchmark has been recently proposed in (Lu et al., 2022a), where the authors provided pixel-based offline datasets collected using varying degrees of a soft actor-critic (SAC) policy. Furthermore, (Lu et al., 2022a) proposed a suite of distractor based datasets with different levels of severity in distractor shift, ranging from easy-shift, medium-shift to hard-shift. In the EASY-EXO setting, we first consider pixel-based offline

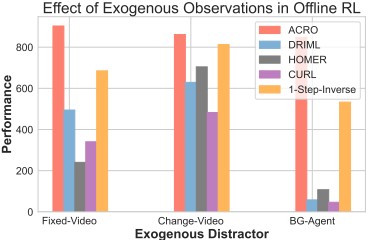 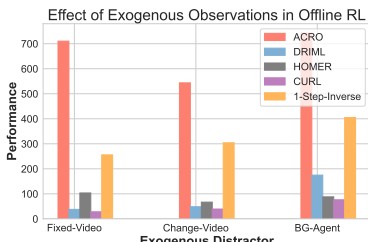

Figure 10: **Summary of Experiment Results on Walker Domain with Expert Dataset:** We vary the type of exogenous distractors present in offline datasets, and evaluate the ability of ACRO for policy learning from provably robust exogenous-free representations, while baseline methods can be prone to the exogenous information present in datasets. From easiest distractor (uncorrelated static images placed in corner or on the side), to corrrelated background exogenous images, and then to fixed or changing video distractors playing in the background, to finally the hardest exogenous information of other random agent information in the agent observation space; we show that ACRO can consistently learn good policies for downstream control tasks, while the ability of baselines to ignore exogenous information dramatically degrades, as we move to hard exogenous information settings. Appendix H.5 provides further ablation studies on the different HARD-EXO offline settings, and performance difference for different domains and datasets.

data without and with visual distractors, as shown in Table 2 and Figure 12 respectively. Experimental results in Table 2 show that even without any exogenous noise, ACRO learns controller latent state representations more accurately than the different state-of-the-art baselines, such that by efficiently decoupling the endogenous state from the exogenous states, policy learning during the offline RL algorithm can lead to significantly better evaluation performance compared to several other baselines.

**Medium-Exogenous Information (MEDIUM-EXO).** We then consider three different types of medium exogenous information that might appear in visual offline data. To that end, we consider exogenous uncorrelated images from STL-10 image dataset (Coates et al., 2011) that appear on the corner or the side of the agent observation, and the goal of ACRO is to filter out the exogenous information while recovering only the controller part of the agent state. We consider *three different types of* MEDIUM-EXO *information:*,

- The exogenous image from STL10 dataset appears in the corner of the agent observation. This does not change the observation size of the agent; and we simply add the exogenous image in one corner, which is fixed during an entire episode during the offline data collection. Figure 13 summarizes the result with different exogenous images placed in the corner. ACRO consistently outperforms several other baselines for a range of different datasets, since it can suitably filter out the exogenous part of the agent state.

- A slightly more difficult setting where now the STL10 exogenous image appears on the side of the agent observation space. This augments the agent observation space from 84 × 84 × 3 to 84 × 84 × 2 × 3 since we consider downsampled STL10 images. Figure 14 summarizes this result comparing ACRO with the baseline representation objectives.

- Finally we consider the distractor setting that has appeared in prior works in online RL (Zhang et al., 2021a) with fixed video distractors playing in the background of agent observation space. For this setting, we specifically re-collect the dataset following the procedure in (Lu et al., 2022a) where the SAC data collecting agent also sees a fixed video distractor playing in the background. Figure 15 summarizes the result and shows performance plots where ACRO can significantly outperform all the baselines across all different types of exogenous datasets.

**Hard Exogenous Information (HARD-EXO).** We finally consider three sets of different hard exogenous information settings, and find that these HARD-EXO can remarkably make it difficult for existing state of the art representation objectives to learnt underlying agent controller states. This setting provides evidence that under suitably constructed exogenous information, which appear highly time correlated during the offline data collection, the baseline methods can fail to capture underly-

ing controller latent representation of states. In contrast, the objective we consider in ACRO, along with the theoretical guarantees for learning a suitable encoder to recover the endogenous states, shows that policy optimization based on the endogenous controller latent states can lead to efficient learning in these control tasks. We consider *three different types of* HARD-EXO *information:*

- We first consider time correlated static images appearing in the background of the agent observation. For this setting, during data collection, the agent sees a fixed image in the background for an entire episode, and it changes per episode of data collection. This time correlated exogenous information ensures that the baselines can remarkably get distracted, while ACRO can still be robust to the static image background. Figure 16 summarizes the results and shows that several existing representation baselines can fail due to time correlated static image distractors.

- We then consider an even more difficult HARD-EXO setting where now the video distractors playing in background also changes per episode during data collection. This is a novel setting with video distractors in RL, since we explicitly consider diverse set of background videos which also changes per episode of data collection. Similar to the above, Figure 17 summarizing the results with diverse video distractors in background per episode, shows that this setting can also break the baseline representation learners to recover the controller latent states, while performance of ACRO remains robust to it, since the ACRO objective learns encoder to recover the endogenous controller latent states accurately.

- Finally, we consider the most challenging HARD-EXO where now in addition to the environment observation, the agent additionally sees other random action agent observations. Here, the goal of the agent is to learn representations to identify the *controllable* environment, while other random-action observations are *uncontrollable* or exogenous to the agent. This is quite a difficult task since the agent we are tring to control also sees observations from the same domain, of other agents playing with random actions. The controller agent observation now consists of other agents placed in a $3 \times 3$ grid. Figure 19 summarizes the experiment results showing that ACRO significantly outperforms all baseline representation learners.

## H.2 EASY-EXO: PIXEL-BASED OFFLINE RL FROM V-D4RL BENCHMARKS

**Visual Offline Control (V-D4RL) without Distractors**. We first verify the effectiveness of learning representations with ACRO without any additional exogenous distractors, and compare with several baselines for learning representations. Figure 11 provides detailed performance curves for Table 2.

**V-D4RL with Varying Severity of Distractor Data Shift**. We then consider the distractor setting in v-d4rl benchmark (Lu et al., 2022a) with varying levels of distractor difficulty. Here, the exogenous noise is based on background static image distractors inducing a distribution shift in the dataset, depending on the level of difficulty from *easy*, to *medium* to *hard* distractors. As shown in Figure 12, we consider two different domains and find that with varying difficulty levels, ACRO can consistently outperform several state of the art baselines, learning directly from pixel data.

## H.3 MEDIUM-EXO: STL10 EXOGENOUS IMAGES OR FIXED VIDEO DISTRACTORS

We extend our experimental results with different types of exogenous image distractors in the observation space of the agent. Detailed description of the dataset collection process is provided in Appendix I.

**Exogenous Image Distractors Placed on the Corner or Side of Agent Observation**. We consider two different settings where the agent environment observation is augmented with STL-10 image Coates et al. (2011) distractors, either placed in the corner or on the side of the environment observation. Here, we consider adding uncorrelated exogenous images where for each pre-training of representations update, the environment observation has image distractors. When placed on the side, it extends the observation space of the agent. Figure 13 and Figure 14 shows results with exogenous images placed in the corner or on the side of the agent observations respectively.

**Fixed Video Distractor**. We first consider a setting where the exogenous distractor in the background is fixed with a single type of video distraction. Figure 15 shows results with fixed video distractor showing that across several datasets, ACRO can consistently outperform baselines.

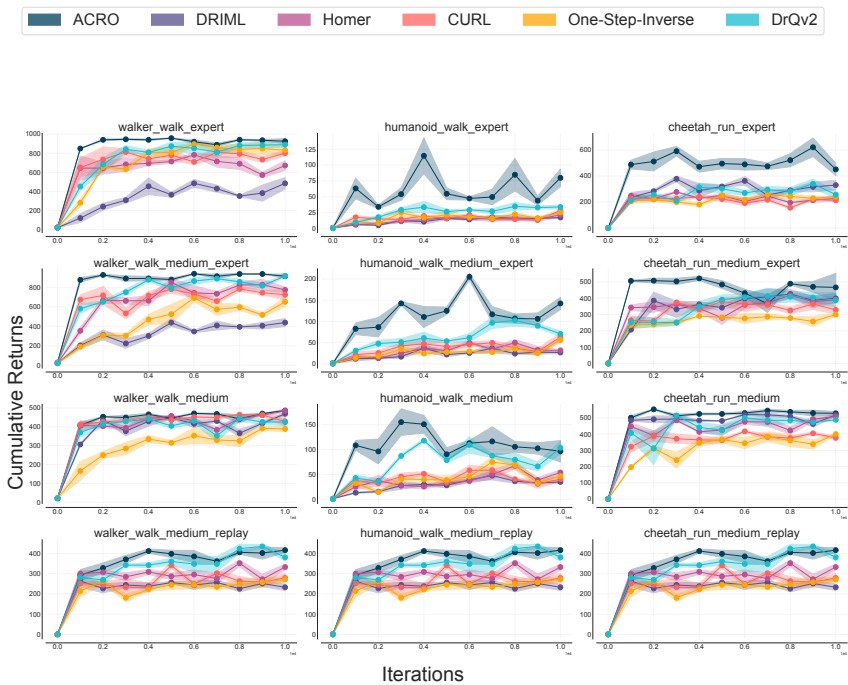

Figure 11: **EASY-EXO-No Distractors Full Results**. Experiments over 6 random seeds. For these experiments, we use the visual offline benchmark from (Lu et al., 2022a) and compare ACRO with several state of the art representation learning objectives. We find that across all tasks, ACRO either outperforms or equally performs as well as the best performing baseline method.

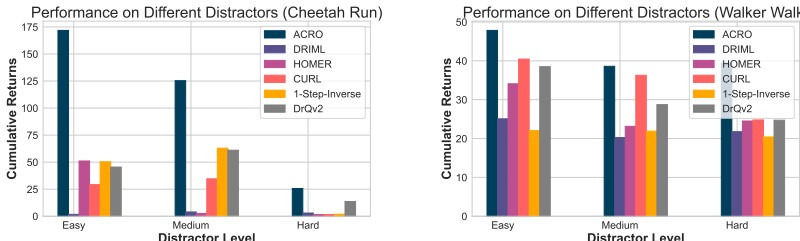

Figure 12: **EASY-EXO-Image Distractors Full Results**. Comparison of ACRO with baselines using the distractor suite of data shift severity from v-d4rl (Lu et al., 2022a) benchmark. We compare results with the two domains and datasets that were released in the v-d4rl benchmark distractor suite.

## H.4 HARD-EXO: TIME CORRELATED AND MOST DIVERSE EXOGENOUS DISTRACTORS

**Static Background Image that Changes Per Episode**. We further experiment with time correlated exogenous distractors in the background, where during every episode of data collection, we provide a background image to the data collecting policy. We find that in presence of exogenous background distractors, ACRO can still be robust to the exogenous noise, while the existing baselines learning representations are more likely to fail, as shown in Figure 16.

**Exogenous Video Distractors that Changes Per Episode**. We then consider a more difficult setting where the type of video distractor playing in background changes during every episode. Further details on the data collection with *fixed* and *changing* video distractor in background is provided in the appendix. Figure 17 shows results with changing video distractor, where ACRO can consistently outperform baselines representation learning methods.

**Multi-Environment Agent Observations as Exogenous Information**. We then consider a setting where the observation space of the agent is augmented with other random agents moving in the

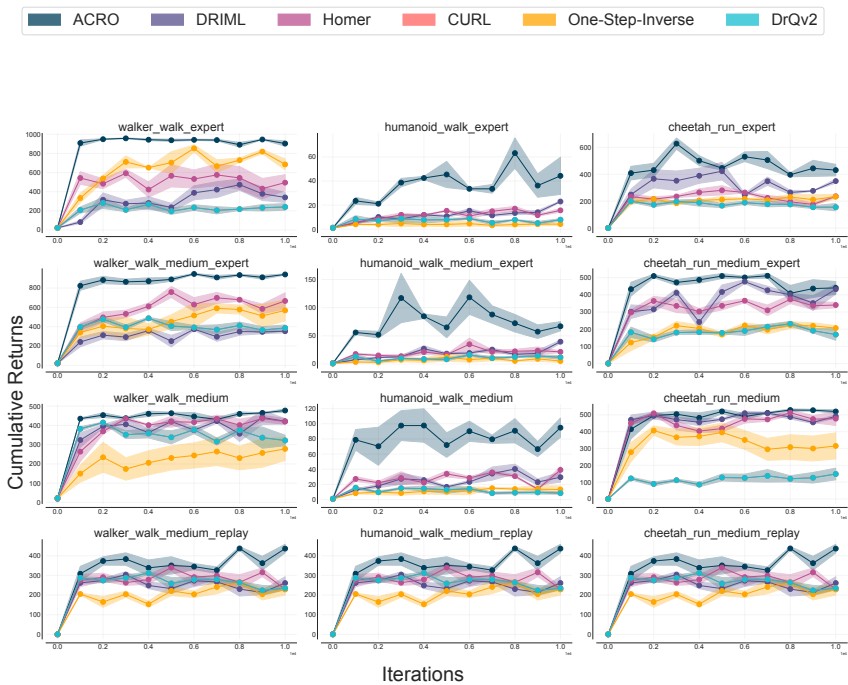

Figure 13: **MEDIUM-EXO-Corner Full Results**. Performance comparison of ACRO with several other baselines.

environment. For this experiment, we take the observations from the environment the agent wants to control, while the other observations of other agents come from a random-policy dataset, either from the same domain, or from different domains. We do this since the random observations from other agents can be treated as exogenous information to the representation encoder; our experiment results show that ACRO can ignore the exogenous information from other agents, leading to significant performance improvements from the offline RL algorithm based on the exogenous observation dataset. Figure 19 summarizes the results.

## H.5 ABLATION STUDIES - HARD-EXO OFFLINE RL

Figure 18 provides a summary of comparison between different datasets in the HARD-EXO noise setting.

## I DATA COLLECTION FOR OFFLINE RL WITH EXOGENOUS INFORMATION

**EASY-EXO Datasets**. For the EASY-EXO setting with exogenous information, we consider uncorrelated visual distractors in the background of the observation space. For this setting, we extensively use the datasets released from the v-d4rl benchmark (Lu et al., 2022a) for offline RL. We note that the data shift severity in v-d4rl benchmark are only limited to two different domains and two data distributions (medium-expert and random). We experiment with both these datasets, and additionally consider the setting with no uncorrelated static images in the background.

**MEDIUM-EXO Datasets**. For the MEDIUM-EXO datasets, we collect new offline datasets using a SAC policy, following the same data collection procedure as in the literature from d4rl benchmark (Fu et al., 2020). The main difference is that when collecting new datasets with the SAC policy, we consider variations of different exogenous noise types in the dataset. As discussed earlier, for the MEDIUM-EXO setting, we collect three different exo-types of datasets : **(a)** Exogenous stl10 images placed in the **corner** of the agent observation space. For this setting, during an episode of data collection, at each time step, we sample a new STL-10 image and place in the corner of the agent observations. **(b)** In the **side** exogenous information setting, instead of the corner, we place

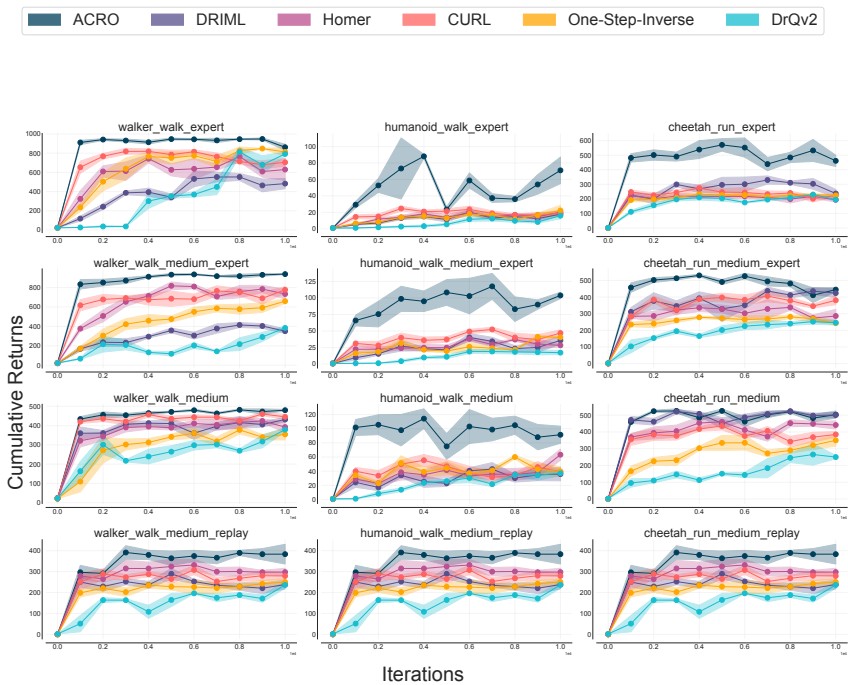

Figure 14: **MEDIUM-EXO-Side Full Results** ACRO can be quite robust to the exogenous images when the exogenous images appear to be similar to the agent in the environment. For example, consider the Cheetah-Run environment with a dog run image on the side, which can be quite distracting to the baseline methods.

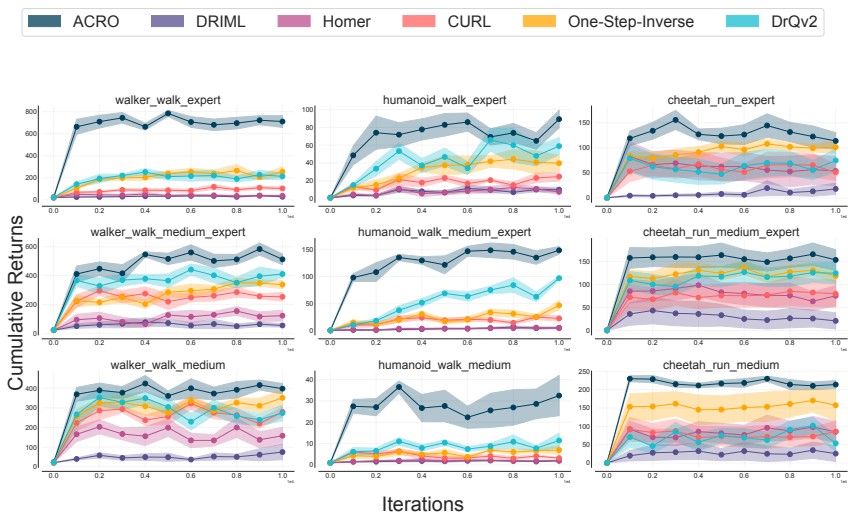

Figure 15: **MEDIUM-EXO-Fixed Video Full Results**. ACRO can outperform baselines with fixed exogenous distractors in background, which is time-correlated in nature. The fixed video distractor setting have often been studied in online RL literature. In this work, we study fixed video distractors in offline RL, where the distractors were present during the offline data collection. .

the exogenous image on the side of the environment observation. This augments the entire agent observaton space. A major difference with (a), however is that, in this setting we consider time correlated exogenous images where each step of SAC policy during an episode sees the same exogenous image, which only changes per episode of data collection. We consider this to be a harder setting compared to changing the exo image at each time-step, since this induced time correlation

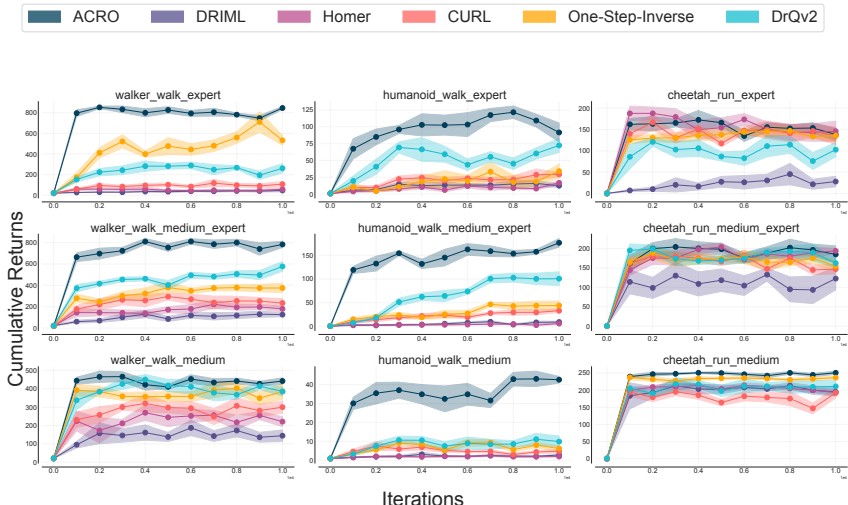

Figure 16: **HARD-EXO-Static Image Full Results**. We find that ACRO can significantly outperform baselines in presence of correlated exogenous static images playing in the background

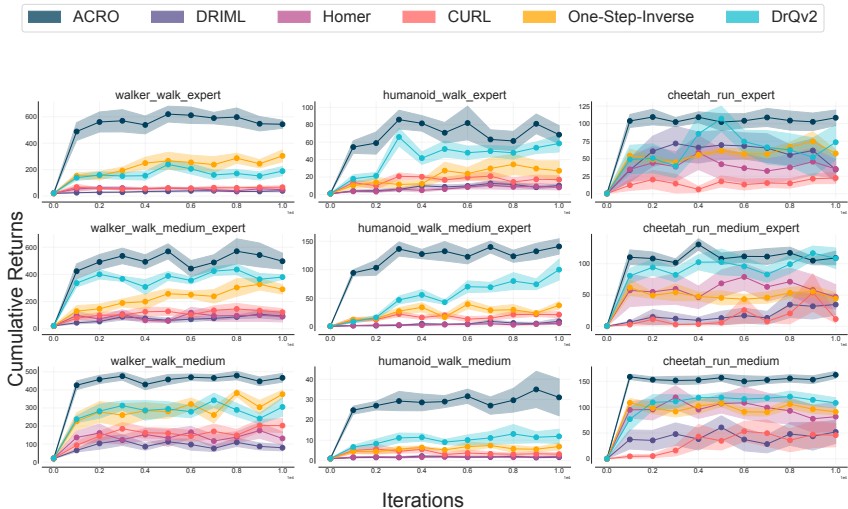

Figure 17: **HARD-EXO-Video Full Results** Across all datasets, ACRO can consistently outperform baseline methods for all types of datasets. When learning representations for offline policy optimization, the ability of ACRO to ignore exogenous information makes it outperform baseine representation objectives in almost all cases. The changing and time correlated video distractors are often hard for baseline methods to ignore, leading to significant performance drops depending on the offline data distribution

can make it harder for the representation objectives to be completely robust to the side exogenous information. **(c)** Finally, we conisder a **fixed video** distractor setting, which has been extensively studied in the online control benchmark (Tassa et al., 2018). Prior works have experimented in the online setting with fixed video distractors. We use the same procedure and fixed video distractor as here, except we use a SAC policy for data collection to be used in the offline setting. All these categories, cumulatively are denoted as MEDIUM-EXO in this work. We release datasets and detailed experiment details for all these settings for MEDIUM-EXO based offline RL datasets.

**HARD-EXO Datasets**. We consider this to be the hardest of the exogenous distractor setting. For this setting, we introduce several new offline benchmarks with different types of exogenous information. **(a) Time Correlated Exogenous Image in the Background** This is a hard distractor type

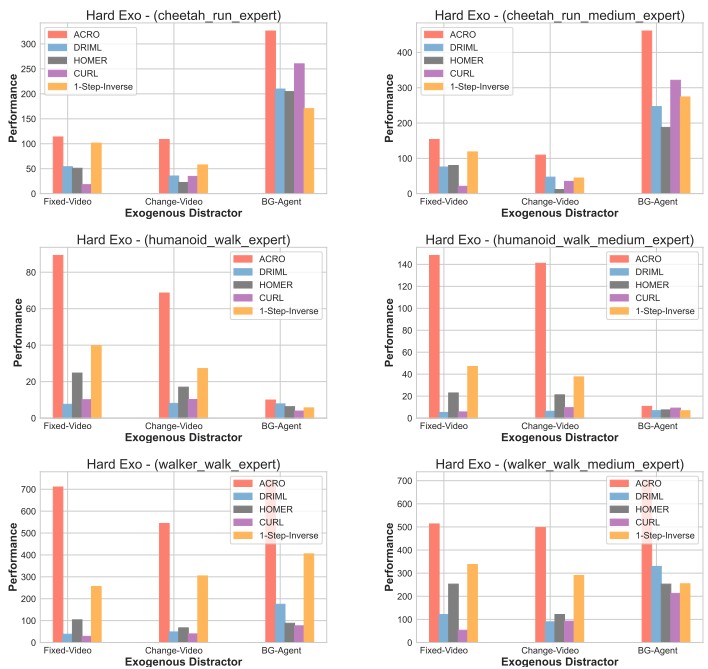

Figure 18: **Summary and Performance Difference between ACRO and baselines in the HARD-EXO offline RL setting**

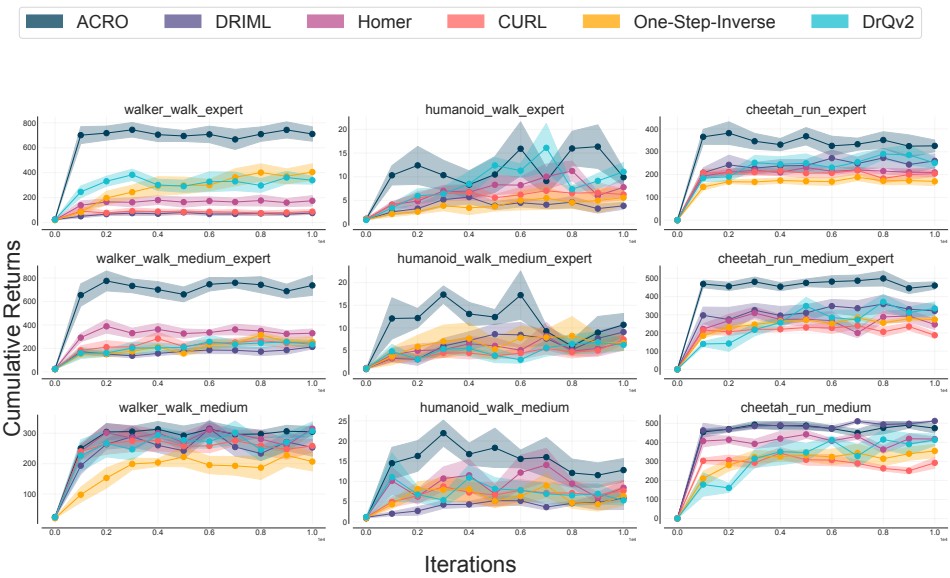

Figure 19: **HARD-EXO-Background Agent Full Results**. Random agent observations placed on the grid, of the entire agent observation space.

where most existing representation learning baselines can fail. For this, during the data collection with SAC agent, we place a static background image from the STL10 dataset in the background. This image remains fixed for all timesteps within an episode, and we only sample a new exogenous background image at every episode. This makes the exogenous noise to be highly time correlated, where baseline representations are likely to capture the background image in addition to learning an embedding of the environment. **(b) Changing Video Distractors** In this setting, like (a), at every new episode we change the *type* of video distractor that we use. During an episode for different

timesteps, the agent already sees correlated noise from the video frames playing in the background. However, since the type of video sequence that plays in background changes, this makes the offline datasets even more difficult to learn from. This setting is inspired by a real world application, where for example, the agent perceiving the world, can see background data from different data distributions (e.g background moving cars compared to people walking in the background). **(c)** Finally, we consider another HARD-EXO dataset, where now during every timestep of data collection, the agent sees other agents playing randomly. Here, we place several other agents, taking random actions, in a grid, and the goal of the agent is to recover only the controllable agent, while being able to ignore the other uncontrollable agents taking random actions, within its observation space. For this setting, the other uncontrollable agents can either be from the same domain (e.g using a Humanoid walker agent for both the endogenous, controllable part and the exogenous, uncontrollable part), or a different setting where the exogenous agents can be from other domains (e.g using a Humanoid agent for the controllable part, while the uncontrollable agents are from a Cheetah agent). We demonstrate both these types of observations either same-exogenous or different-exogenous in Figure 21 and Figure 20 respectively. We also release these datasets as an additional contribution to this work, and hope that future works in offline RL will use these datasets as benchmarks, for learning robust representations.

## J  BACKGROUND AGENT CONSECUTIVE FRAMES VISUALIZATION

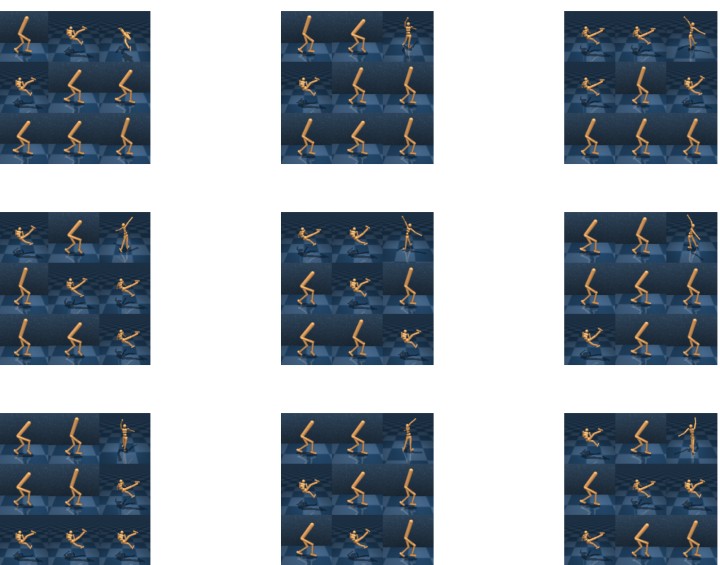

Figure 20: Consecutive timesteps from three different episodes (each row); where the controllable agent and the background agents are from different domains

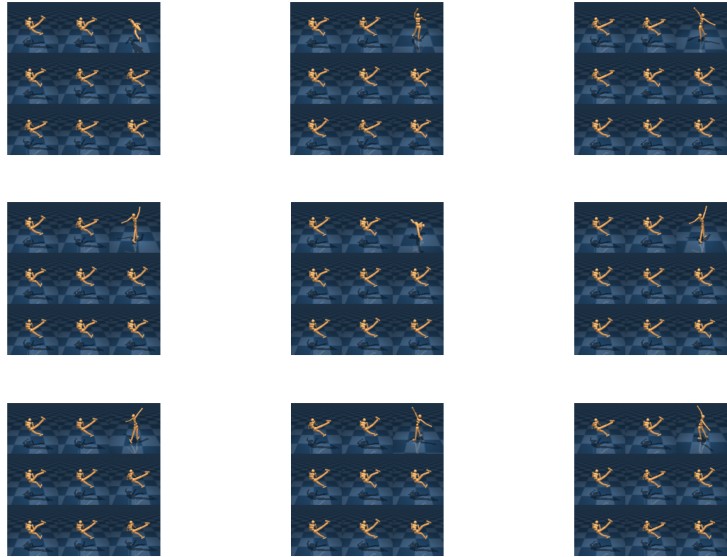

Figure 21: Consecutive timesteps from three different episodes (each row); where the controllable agent and the background agents are from same domains

## K  VISUALIZING RECONSTRUCTIONS FROM THE DECODER

We show additional reconstructions for ACRO, DrQ, behavior cloning, and random features on different types of exogenous noise in Figure 22, Figure 23, and Figure 24.

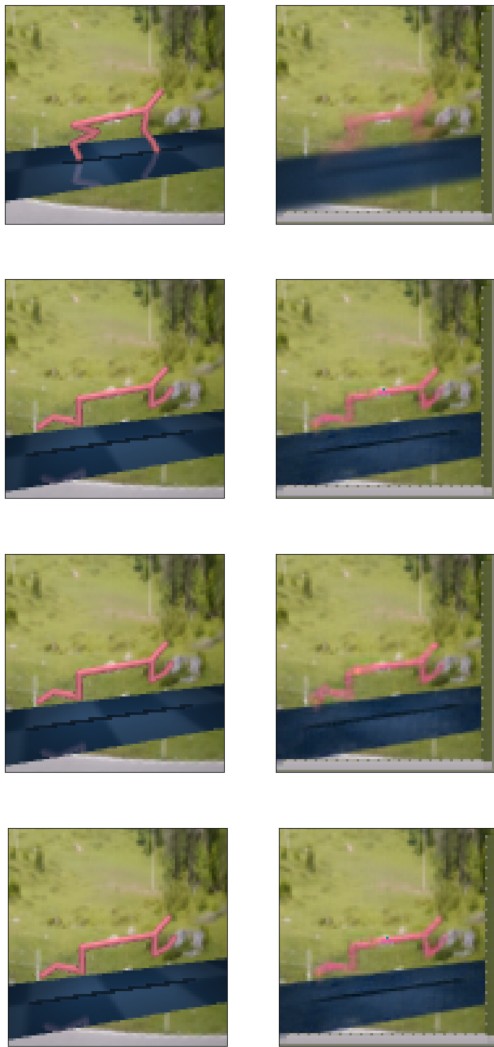

Figure 22: **Reconstructions for Fixed Background Distractor** from a decoder learnt of over two kinds of representations: **Top-Bottom**: Random features, DrQ, Behavior cloning, and ACRO. **Left Column**: Original observation, **Right column**: Reconstruction.

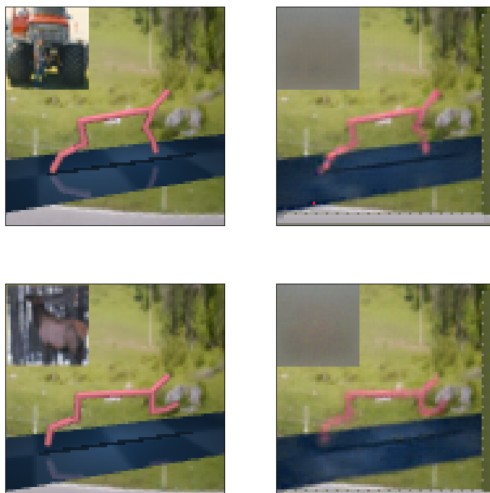

Figure 23: **Reconstructions with CIFAR images in background** from a decoder learnt of two kinds of representations: **Top-Bottom**: ACRO, DrQ. **Left Column**: Original observation, **Right column**: Reconstruction.

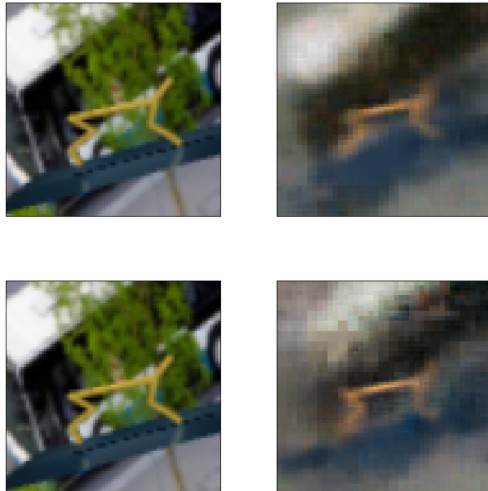

Figure 24: **Reconstructions with a distractor video** playing in the background from a decoder learnt of two kinds of representations: **Top-Bottom**: BC, ACRO. **Left Column**: Original observation, **Right column**: Reconstruction.

