# OpenReview forum: "Agent-Controller Representations: Principled Offline RL with Rich Exogenous Information"
_NeurIPS.cc/2022/Workshop/Offline_RL — Offline RL Workshop NeurIPS 2022_

### Official Review · Reviewer_dVgj · 2022-10-18
**Good paper with some additional evaluation suggestions**

**Rating:** 7
**Confidence:** 2

**Review:**

Summary:
The paper presents a method for learning representations from offline RL datasets based on inverse dynamics models in order to mitigate the effect of exogenous visual information when training robotic policies. In particular, the paper defines a training target for predicting the first action given the current state and the next or some multi-step future state. After pre-training on such prediction, the model can be fine-tuned on a given RL task. The paper shows that exogenous non-important visual information can be harmful for RL and that the presented method effectively learns representation that are invariant to it. The method is shown to work on simulated tasks with varying degrees of exogenous information.

Strengths:
- Exogenous information can make training a robotic policy slow and inefficient, effective pre-training techniques could help to train robotic policies on more realistic scenarios.
- The method is self-supervised and hence can be easily scaled to very large pre-training datasets.
- The paper is well written and easy to understand and follow.
- The method is shown to perform well with various amounts of exogenous information.

Criticism:
- It would be useful to see a comparison to other common pre-training techniques, e.g. just using pre-trained visual features such as ResNet, or pre-training using next-state prediction (forward model) or goal-conditioned objectives.
- It would be interesting to see a wider range of robotic tasks in the evaluation.